

# Prognostic value of complementary biomarkers of neurodegeneration in a mixed memory clinic cohort

Mathias Holsey Gramkow[1], Le Gjerum[1], Juha Koikkalainen[2], Jyrki Lötjönen[2], Ian Law[3], Steen Gregers Hasselbalch[1], Gunhild Waldemar[1] and Kristian Steen Frederiksen[1]

[1] Danish Dementia Research Centre, Department of Neurology, Rigshospitalet, University of Copenhagen, Copenhagen, Denmark
[2] Combinostics Ltd., Tampere, Finland
[3] Department of Clinical Physiology, Nuclear Medicine and PET, Rigshospitalet, University of Copenhagen, Copenhagen, Denmark

Corresponding author
Mathias Holsey Gramkow,
mathias.holsey.gramkow@regionh.dk

## ABSTRACT

**Background**. Biomarkers of neurodegeneration, e.g. MRI brain atrophy and [18F]FDG-PET hypometabolism, are often evaluated in patients suspected of neurodegenerative disease.

**Objective**. Our primary objective was to investigate prognostic properties of atrophy and hypometabolism.

**Methods**. From March 2015-June 2016, 149 patients referred to a university hospital memory clinic were included. The primary outcome was progression/stable disease course as assessed by a clinician at 12 months follow-up. Intracohort defined $z$-scores of baseline MRI automatic quantified volume and [18F]FDG-PET standardized uptake value ratios were calculated for all unilaterally defined brain lobes and dichotomized as pronounced atrophy (+A)/ pronounced hypometabolism (+H) at $z$-score $<0$. A logistic regression model with progression status as the outcome was carried out with number of lobes with the patterns +A/-H, -A/+H, +A/+H respectively as predictors. The model was mutually adjusted along with adjustment for age and sex. A sensitivity analysis with a $z$-score dichotomization at $-0.1$ and $-0.5$ and dichotomization regarding number of lobes affected at one and three lobes was done.

**Results**. Median follow-up time was 420 days [IQR: 387-461 days] and 50 patients progressed. Patients with two or more lobes affected by the pattern +A/+H compared to patients with 0–1 lobes affected had a statistically significant increased risk of progression (odds ratio, 95 % confidence interval: 4.33, 1.90–9.86) in a multivariable model. The model was partially robust to the applied sensitivity analysis.

**Conclusion**. Combined atrophy and hypometabolism as assessed by MRI and [18F]FDG-PET in patients under suspicion of neurodegenerative disease predicts progression over 1 year.

## INTRODUCTION

Dementia is a substantial health problem worldwide as disease incidence increases with advancing age and the global older generation is growing with an increasing rate according to the UN World Population Prospect from 2019. Forty-seven million patients were estimated to live with dementia in 2015 and it is forecast that dementia will affect 132 million individuals by 2050 (*Geneva: World Health Organization, 2017*). Neurodegeneration is a disease state defined by progressive loss of neuronal function and structure (*Spillantini & Goedert, 2013*). Many dementia disorders, the most prominent being Alzheimer's disease (AD), frontotemporal dementia and dementia with Lewy bodies, share this neuropathological property (*Prince et al., 2013*; *Masters et al., 2015*; *Mueller et al., 2017*; *Bang, Spina & Miller, 2015*).

Use of biomarkers reflecting neurodegeneration may be a part of the clinical work-up when evaluating patients suspected of cognitive impairment (*Albert et al., 2011*). Apart from helping to establish the diagnosis, neurodegenerative biomarkers may predict disease course. Established biomarkers of neurodegenerative diseases are tau protein measured in the cerebrospinal fluid (CSF), brain atrophy assessed by MRI and 2-[$^{18}$F]fluoro-2-deoxy-D-glucose ([$^{18}$F]FDG) PET assessment of brain hypometabolism (*Jack et al., 2013*). While prognostic properties of single markers have previously been evaluated (*Chetelat et al., 2005*; *Perani et al., 2016*; *Ottoy et al., 2019*), the combination of markers has not been tested extensively. Since two or more biomarkers are often available to the clinician for the same patient, performance of combined biomarkers may be more useful and represents a more clinically oriented way of viewing biomarkers. Physicians are often faced with conflicting results when looking at a plethora of markers. This makes interpretation in clinical practice difficult with regards to diagnosis but also when assessing risk for progression (*McKhann et al., 2011*), and validated biomarkers of progression are effectively not available to clinicians (*McGhee et al., 2014*). Moreover, while MRI and [$^{18}$F]FDG -PET are measures of different aspects in the neurodegenerative process, they nevertheless are markers of the same process. Therefore, it remains undetermined whether the 2 modalities hold complementary or additive information with regards to risk of progression.

An evaluation and classification of the existing biomarkers of neurodegeneration in order to identify the most appropriate markers to predict progression will aid the clinicians to initiate relevant care and recognize potential patients for early therapeutic intervention. In addition to this, patients may wish to know the likely course of their disease, including risk of progression, in order to plan for the future. This study will evaluate MRI brain atrophy, [$^{18}$F]FDG-PET and CSF-total-tau and their usefulness in assessing risk of clinical progression in patients referred to a memory clinic under suspicion of neurodegenerative disease. We hypothesized that brain atrophy, hypometabolism and high CSF-total tau were associated with clinical progression in patients suspected of neurodegenerative disease. Secondly, we hypothesized that there was congruency between biomarkers of neurodegeneration with regards to progression.

## MATERIALS & METHODS

### Participants, study protocol and ethical statement

A total of 149 patients referred to the Memory clinic at Rigshospitalet (RH) in Copenhagen, Denmark suspected of neurodegenerative disease were included in the present study. The cohort was part of a larger EU-supported study, PredictND, which investigated the clinical impact of a computer assisted decision support tool in diagnosis and prognosis of patients referred to memory clinics with suspicion of neurodegenerative disease. The original study recruited 208 patients at RH. The study details of PredictND are reported elsewhere (*Bruun et al., 2019*). In short, the inclusion criteria for the PredictND study were (1) patients suspected of having cognitive complaints as a result of subjective cognitive decline (SCD), mild cognitive impairment (MCI) or dementia (2) a Mini-Mental State Examination (MMSE) $\geq$ 18 and a T1-weighted MRI ($\geq$1.5 T) available. Exclusion criteria were major psychiatric disorder, excessive alcohol intake or substance abuse within the last two years and other brain disorders that could explain the cognitive complaints. Inclusion criteria for the present study were (1) recruited at RH (2) a brain MRI at baseline including a T1-weighted MR image with slice thickness <two mm of sufficient quality for analysis, (3) [$^{18}$F]FDG -PET at baseline, (4) at least 12-months follow-up. Patients in the PredictND study were evaluated at a consensus conference held for specialist physicians, nurses and neuropsychologist at the respective clinics and a diagnosis was given, the criteria for which are described elsewhere (*Bruun et al., 2019*). The PredictND study was approved by the Scientific Ethics Committee of the Capital Region of Denmark (H-1-2014-126) and study experiments were carried out in accordance with the Helsinki Declaration. All patients provided written informed consent for their data to be used for research purposes.

### Outcome assessment

The primary outcome for the present study was progression assessed at a follow-up clinical evaluation 12 months post-diagnosis. The 12-month follow-up clinical evaluation was done by an experienced dementia specialist, however not necessarily the same clinician who did the initial consultation. At the follow-up visit, where the patient was evaluated with at least a Mini Mental State Examination (MMSE) (*Folstein, Folstein & McHugh, 1975*) and a Clinical Dementia Rating (*Hughes et al., 1982*), the disease course was determined as either progression, fluctuation, stable and improvement. Baseline scans and results of the CSF analysis were not included in the assessment of disease course at 12-month follow-up, although the clinician was not blinded to diagnostic test data. In the present study, the disease course categories fluctuation, improvement and stable were collapsed into a single category called stable, as too few patients' disease courses were categorized as fluctuation (one patient) or improvement (five patients) for a meaningful analysis to be undertaken.

### MRI

Scans were acquired using a T1-weighted gradient echo sequence on 3 T scanners. Automated image quantification analysis was done in the PredictND tool as described by Koikkalainen et al. (*Koikkalainen et al., 2016*). In short, 3-D images were segmented using a multi-atlas segmentation approach to 139 regions. Regions of interest (ROIs) for

individual bilaterally defined brain lobes (occipital, temporal, frontal and parietal) and a total ventricular volume were constructed from this initial segmentation of structures listed under Table S1. The total ventricular volume was chosen as a possible prognostic marker, as it had shown promise within AD (*McGhee et al., 2014*).

## [$^{18}$F]FDG -PET

[$^{18}$F]FDG -PET scans were acquired using either GE Medical, Philips, or Siemens PET scanners according to international practice guidelines (*Varrone et al., 2009*). An in-house developed software tool was used for the data analysis of images. Images were co-registered to their corresponding segmented MR images to generate corresponding ROIs as described under *MRI*, and an ROI of total grey and white matter uptake was defined. Standardized uptake value ratios (SUVRs) for ROIs were referenced to the mean activity in the white matter of the cerebellum.

## CSF-total tau

CSF was collected by lumbar puncture at the baseline visit and handled according to standard operating procedures. CSF-total tau was measured using a commercially available enzyme-linked immunosorbent assay (Innotest, Fujirebio, Ghent, Belgium). In total, 76 patients had CSF-total tau measured.

## Statistical analysis

Spearman's $\rho$ was calculated for baseline variables of interest. Median follow-up time was defined as days from baseline visit till follow-up visit. $Z$-scores for the biomarkers MRI-total ventricular volume, [$^{18}$F]FDG -PET-total grey and white matter SUVRs and CSF-total tau were calculated using the intracohort mean and standard deviation for each variable. Variables were dichotomized and categorized as abnormal if the $z$-score was higher (MRI-total ventricular volume and CSF-total tau) or lower ([$^{18}$F]FDG -PET-total grey and white matter SUVRs) than zero. The number of abnormal markers were used as a predictor in a multivariable logistic regression model (M1) with progression status (progressed/stable) at follow-up as the outcome. Interval validation was done in the form of 10-fold 1000 times cross validation using the *caret* package in R and balanced accuracy for the model was calculated. $Z$-scores were calculated in the same way for the MRI intralobal volumes and [$^{18}$F]FDG -PET SUVRs for each unilaterally defined lobe (left and right frontal, parietal, occipital and temporal). Dichotomization for these variables were done at a $z$-score = 0 and each lobe was defined with the labels pronounced atrophy ($z$-score for MRI volume < 0) or unpronounced atrophy ($z$-score for MRI volume > 0) and concurrently pronounced hypometabolism ($z$-score for [$^{18}$F]FDG -PET SUVR < 0) or unpronounced hypometabolism ($z$-score for [$^{18}$F]FDG -PET SUVR > 0). An atrophy/hypometabolism pattern for each lobe was defined by either pronounced atrophy (+A) or unpronounced atrophy (-A), and pronounced hypometabolism (+H) or unpronounced hypometabolism (-H). Three variables, each counting the number of lobes with the patterns +A/-H, -A/+H and +A/+H respectively were then entered as separate dichotomous predictors in a multivariable logistic regression model (M2) with the same outcome and adjustment as in M1 as well as mutual adjustment. Lastly, anatomical variables were constructed with

levels explained and illustrated in Fig. S1. Shortly, levels for each lobe (left and right lobe taken as a whole) were defined as (1) isolated atrophy, (2) isolated hypometabolism, (3) congruent atrophy and hypometabolism and non-isolated atrophy or hypometabolism (4) no abnormality. Anatomical variables were added as predictors in M2 in an exploratory model (M3) and mutually adjusted for in a multivariable model. Multicollinearity in M3 was tested for by eigensystem analysis. Sensitivity analyses, investigating dichotomization at a $z$-score of 0.1 and 0.5 (MRI-total ventricular volume and CSF-total tau) and at $-0.1$ and $-0.5$ ([$^{18}$F]FDG -PET-total grey and white matter SUVRs, pronounced/unpronounced atrophy and pronounced /unpronounced hypometabolism), were carried out in M1 and M2. Furthermore, a sensitivity analysis was carried out in M2 with dichotomization at 1 lobe affected and 3 lobes affected by specific patterns. The same models were run (M2 and M3) as pre-planned analyses in a sub-cohort of patients with MCI and dementia. Linear regression was done for predictors in M1 against MMSE total score follow-up score subtracted from baseline and linear regression model assumptions were checked by visual inspection of Q-Q and residuals plots. For all constructed logistic regression models, odds ratios (ORs) and 95% confidence intervals (95% CIs) were calculated using the *finalfit* package. Only two-sided tests were used and a significance level of 0.05 was imposed. All statistical analyses were carried out in R (3.2.3) (*Team, 2016*).

# RESULTS

The baseline characteristics of patients included in the present study are presented in Table 1. The mean age was 70.3 years (SD 9.8) and males and females were near equally represented in the material. Patients' median MMSE total score was 28 (range 18–30). Median follow-up time was 420 days [IQR: 387–461 days] and at follow-up 50 patients had progressed. Figure 1 shows the Spearman correlations between variables of interest. MRI total ventricular volume correlated negatively with total grey and white matter uptake [$^{18}$F]FDG -PET with ($\rho = -0.25$) and positively with age ($\rho = 0.38$). MMSE total score correlated positively with [$^{18}$F]FDG -PET total grey and white matter uptake ($\rho = 0.31$) and negatively with CSF-total tau ($\rho = -0.28$). All other correlations were not significant ($\rho$ not shown). Linear regression of each biomarker of neurodegeneration (MRI-total ventricular volume, [$^{18}$F]FDG -PET total grey and white matter uptake and CSF-total tau) against the MMSE score difference between follow-up and baseline are shown in the Figs. S2–S4. In these models, [$^{18}$F]FDG -PET total grey and white matter uptake associated positively with the MMSE change from baseline to follow-up ($r^2 = 0.039$, $p = 0.017$).

## Abnormal markers of neurodegeneration and progression

Table 2 gives the results of a logistic regression analysis with biomarkers of neurodegeneration (MRI total ventricular volume, [$^{18}$F]FDG -PET total grey and white matter uptake and CSF-total tau) as predictors of progression at follow-up. The results of Model 1 (Table 2) showed that with each additional abnormal marker the risk of progression increased when tested against zero abnormal markers (1 abnormal marker: OR 2.38, 95% CI [0.59–9.66], 2 abnormal markers: OR 2.40, 95% CI [0.59–9.71], and 3 abnormal markers: OR 3.66, 95% CI [0.51–26.36], in the multivariable model). Model 1

**Table 1  Baseline characteristics.**

| Variable | Progression (N = 50) | Stable (N = 99) | Total (N = 149) | P-value |
|---|---|---|---|---|
| **Gender** | | | | 0.687[1] |
| Female | 27 (54.0%) | 50 (50.5%) | 77 (51.7%) | |
| Male | 23 (46.0%) | 49 (49.5%) | 72 (48.3%) | |
| **Age, years** | | | | 0.003[2] |
| Mean (SD) | 73.6 (9.2) | 68.6 (9.7) | 70.3 (9.8) | |
| **Mini-Mental State Examination total score** | | | | <0.001[3] |
| Median (range) | 26 (19–30) | 29 (18–30) | 28 (18–30) | |
| **Consensus conference diagnosis** | | | | <0.001[4] |
| Dementia | 42 (84.0%) | 39 (39.4%) | 81 (54.4%) | |
| Mild cognitive impairment (MCI) | 7 (14.0%) | 26 (26.3%) | 33 (22.1%) | |
| Subjective cognitive decline | 1 (2.0%) | 34 (34.3%) | 35 (23.5%) | |
| **Dementia etiology (consensus conference diagnosis)** | | | | 0.308[4] |
| Alzheimer's dementia | 28 (66.7%) | 19 (48.7%) | 47 (58.0%) | |
| Atypical Alzheimer's dementia | 1 (2.4%) | 2 (5.1%) | 3 (3.7%) | |
| Atypical parkinsonism + Parkinson's disease with dementia | 2 (4.8%) | 1 (2.6%) | 3 (3.7%) | |
| Alcohol-related dementia | 0 (0.0%) | 1 (2.6%) | 1 (1.2%) | |
| Frontotemporal dementia | 2 (4.8%) | 1 (2.6%) | 3 (3.7%) | |
| Lewy body dementia | 0 (0.0%) | 4 (10.3%) | 4 (4.9%) | |
| Mixed dementia | 1 (2.4%) | 3 (7.7%) | 4 (4.9%) | |
| Normal pressure hydrocephalus | 0 (0.0%) | 1 (2.6%) | 1 (1.2%) | |
| Other | 2 (4.8%) | 3 (7.7%) | 5 (6.2%) | |
| Vascular dementia | 6 (14.3%) | 4 (10.3%) | 10 (12.3%) | |
| **CSF-total tau (ng/L)** | | | | 0.259[3] |
| Median (range), $n = 76$ | 370 (126–1200) | 347 (36–993) | 360 (36–1200) | |
| N (%) with elevated (>400 ng/L) tau | 21 (42.9%) | 13 (48.1%) | 34 (44.7%) | |
| **MRI total ventricular volume (mm$^3$)** | | | | 0.007[3] |
| Median (range) | 64772 (24318–239803) | 50402 (14760–193975) | 57363 (14760–239803) | |
| **$^{18}$F FDG-PET Cerebral gray and white matter total uptake (SUVr normalized to cerebellum)** | | | | <0.001[2] |
| Mean (SD) | 1.050 (0.081) | 1.113 (0.112) | 1.092 (0.106) | |

**Notes.**

Tests for differences between stable and progressed were as follows: [1]Pearson's Chi-squared test, [2]ANOVA, [3]Kruskal–Wallis rank sum test, [4]Fisher's Exact Test.

had an accuracy of 0.59 in a 10-fold 1000 times cross-validation and a balanced accuracy of 0.53 in a simple prediction of progression within the dataset. Directionality of point estimates were only partially robust to sensitivity analyses (Table S2). Model 2 (Table 2) did not include CSF-total tau as a predictor but was otherwise identical to Model 1. The results of Model 2 showed that with each additional abnormal marker the risk of progression increased when tested against zero abnormal markers (1 abnormal marker: OR 2.68, 95% CI [1.01–7.12] and 2 abnormal markers: OR 3.45, 95% CI [1.16–10.28], in the multivariable model). Model 2 had an accuracy of 0.63 in a 10-fold 1000 times cross-validation and a

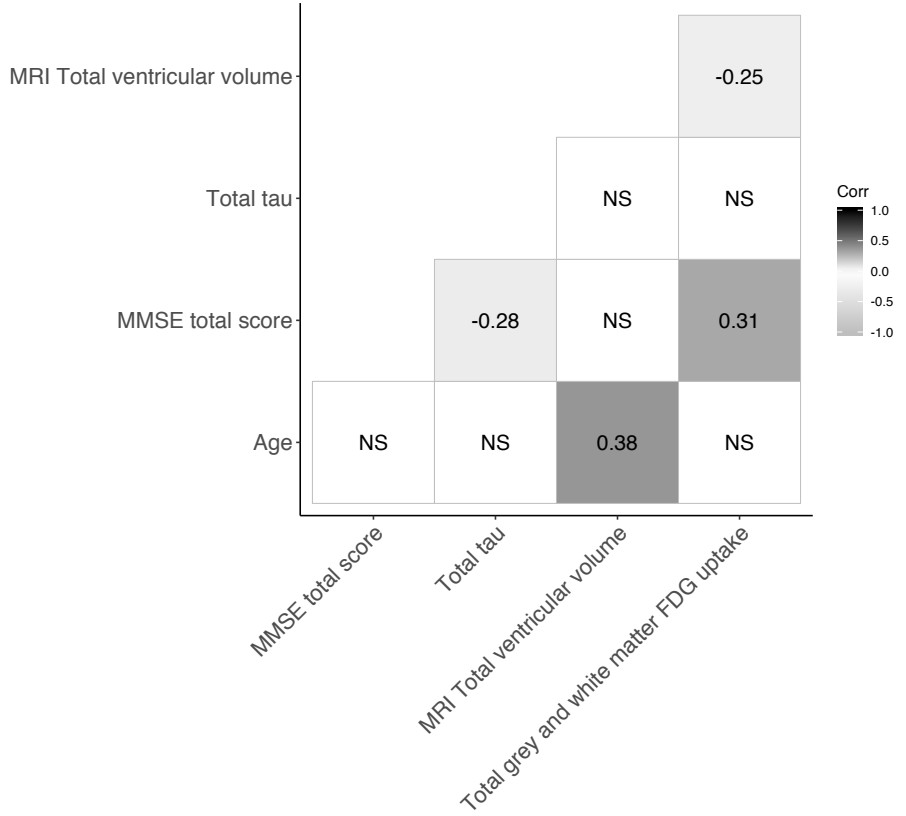

**Figure 1** **Correlogram showing Spearman correlations of markers of neurodegeneration.** For subset of cohort with CSF-total tau, N=76. NS = not significant.

balanced accuracy of 0.58. Directionality and 95% CIs for point estimates were robust to a $z$-score dichotomization at $-0.1/0.1$, but not at $-0.5/0.5$ (Table S2).

## Atrophy and hypometabolism patterns and progression

Table 3 gives the results of a logistic regression analysis investigating variables expressing different patterns of atrophy (+A) and hypometabolism (+H) as predictors of progression. Comparing patients who had two or more lobes affected by the pattern +A and +H with patients who had 0–1 affected lobes resulted in an OR of 4.46 (95% CI [2.16–9.22]) in the univariable model, and an OR of 4.33 (95% CI [1.90–9.86]) in the multivariable model. This indicated an overall four-fold increase in the odds of 1-year progression, which reached statistical significance ($p < 0.001$). The directionality and statistical significance of this point estimate was robust to a sensitivity analysis investigating a dichotomization at a $z$-score value of $-0.1$. The same was true when investigating a dichotomization at 1 or more affected lobes and 3 or more affected lobes with a $z$-score dichotomization kept at 0, meaning that an increased risk of progression was observed for patients with 1 or more and 3 or more lobes affected using the same comparison as above. A sensitivity analysis investigating a $z$-score dichotomization at $-0.5$ showed a preservation of directionality, although statistical significance was not reached for the multivariable estimate (Table S2).

**Table 2** Markers of neurodegeneration and their relation to clinical progression.

| Variable | Levels | Stable | Progression | Odds ratio (univariable) | Odds ratio (multivariable) |
|---|---|---|---|---|---|
| **Model 1** | | | | | |
| Age | Mean (SD) | 68.7 (8.9)[b] | 71.1 (9.9)[c] | 1.03 (0.98–1.09, $p = 0.272$) | 1.02 (0.96–1.08, $p = 0.497$) |
| Gender | Female | 24 (49.0) | 13 (48.1) | – | – |
| | Male | 25 (51.0) | 14 (51.9) | 1.03 (0.40–2.65, $p = 0.945$) | 1.01 (0.38–2.70, $p = 0.983$) |
| Abnormal markers (MRI, [$^{18}$F]FDG-PET and tau) [a] | 0 | 16 (32.7) | 4 (14.8) | – | – |
| | 1 | 14 (28.6) | 9 (33.3) | 2.57 (0.65–10.21, $p = 0.179$) | 2.38 (0.59–9.66, $p = 0.225$) |
| | 2 | 16 (32.7) | 11 (40.7) | 2.75 (0.72–10.48, $p = 0.138$) | 2.40 (0.59–9.71, $p = 0.221$) |
| | 3 | 3 (6.1) | 3 (11.1) | 4.00 (0.58–27.82, $p = 0.161$) | 3.66 (0.51–26.36, $p = 0.198$) |
| **Model 2** | | | | | |
| Age | Mean (SD) | 68.6 (9.7) | 73.6 (9.2) | 1.06 (1.02–1.10, $p = 0.004$) | 1.04 (0.99–1.08, $p = 0.103$) |
| Gender | Female | 50 (50.5) | 27 (54.0) | – | – |
| | Male | 49 (49.5) | 23 (46.0) | 0.87 (0.44–1.72, $p = 0.687$) | 0.78 (0.37–1.65, $p = 0.517$) |
| Abnormal markers (MRI, [$^{18}$F]FDG-PET) | 0 | 42 (42.4) | 8 (16.0) | – | – |
| | 1 | 35 (35.4) | 23 (46.0) | 3.45 (1.37–8.67, $p = 0.008$) | 2.68 (1.01–7.12, $p = 0.048$) |
| | 2 | 22 (22.2) | 19 (38.0) | 4.53 (1.71–12.01, $p = 0.002$) | 3.45 (1.16–10.28, $p = 0.026$) |

**Notes.**

[a] +/- refers to a $z$-score$< 0$ in an affected lobe (right and left hemisphere frontal, temporal, parietal and/or occipital) for either hypometabolism ([$^{18}$F]FDG-PET uptake) and/or atrophy (MRI volume).

[b] n (% of stable) if nothing else stated under level.

[c] n (% of progressed) if nothing else stated under level.

**Table 3** Atrophy/hypometabolism patterns and their relation to clinical progression.

| Variable | Level | Stable | Progressed | Odds ratio (univariable) | Odds ratio (multivariable) |
|---|---|---|---|---|---|
| Age | Mean (SD) | 68.6 (9.7)[b] | 73.6 (9.2)[c] | 1.06 (1.02-1.10, $p = 0.004$) | 1.04 (1.00-1.09, $p = 0.045$) |
| Sex | Female | 50 (50.5) | 27 (54.0) | – | – |
| | Male | 49 (49.5) | 23 (46.0) | 0.87 (0.44-1.72, $p = 0.687$) | 1.48 (0.59-3.71, $p = 0.406$) |
| +Atrophy, -Hypometabolism[a] | 0-1 affected lobes | 58 (58.6) | 32 (64.0) | – | – |
| | 2 or more affected lobes | 41 (41.4) | 18 (36.0) | 0.80 (0.39-1.61, $p = 0.524$) | 1.15 (0.50-2.65, $p = 0.742$) |
| -Atrophy, +Hypometabolism | 0-1 affected lobes | 67 (67.7) | 32 (64.0) | – | – |
| | 2 or more affected lobes | 32 (32.3) | 18 (36.0) | 1.18 (0.58-2.41, $p = 0.654$) | 1.14 (0.45-2.90, $p = 0.778$) |
| +Atrophy, +Hypometabolism | 0-1 affected lobes | 69 (69.7) | 17 (34.0) | – | – |
| | 2 or more affected lobes | 30 (30.3) | 33 (66.0) | 4.46 (2.16-9.22, $p < 0.001$) | 4.33 (1.90-9.86, $p < 0.001$) |

**Notes.**

[a] +/- refers to a $z$-score$< 0$ in an affected lobe (right and left hemisphere frontal, temporal, parietal and/or occipital) for either hypometabolism ([$^{18}$F]FDG-PET uptake) and/or atrophy (MRI volume).

[b] (% of stable) if nothing else stated under level.

[c] n (% of progressed) if nothing else stated under level.

To explore our chosen outcome of physician's assessment's validity, we also evaluated a model using the outcome of a decline of 3 or more points on the MMSE scale as an indication of progression (see Table S3A), which did not remarkably change the results of the model presented in Table 3. We also explored the influence of disease duration and educational level of the participants as possible confounders in an additional model (see

Table S3B), where we included these variables to the model presented in Table 3. This analysis showed no discernible influence on the results obtained.

A pre-planned subgroup analysis with parameters identical to the model pertaining to the results reported in Table 3 was carried out in patients diagnosed with MCI or dementia, thus excluding patients with SCD (Table S4). We found that having 2 or more lobes affected by the pattern +A and +H was associated with an increased probability of progression (univariable: OR 2.82, 95% CI [1.31–6.10]; multivariable: OR 2.91, 95% CI [1.22–6.97]).

## Anatomical variation of atrophy and hypometabolism patterns and progression

Table 4 gives the results of an exploratory logistic regression analysis investigating anatomical variation in patterns of atrophy and hypometabolism as predictors of progression (For detailed illustration of the construction of variables see Fig. S4). The results of the multivariable analysis showed that the presence of intralobal congruent atrophy and hypometabolism and/or the presence of atrophy/hypometabolism that was not isolated in the occipital lobe was associated with a lower probability of progression compared to having no abnormality (OR 0.06, 95% CI [0.01–0.46]). Eigensystem analysis resulted in a condition number of 10.3, indicating little or no concern of multicollinearity in this model.

A pre-planned subgroup analysis with parameters identical to the model reported in Table 4 was carried out in patients diagnosed with MCI or dementia, thus excluding patients with SCD. The results of this analysis are shown in Table S5. The results of this multivariable analysis showed that the presence of intralobal congruent atrophy and hypometabolism and/or non-isolated atrophy/hypometabolism in the occipital lobe was associated with a lower probability of progression compared to having no abnormality (OR 0.16, 95% CI [0.02–1.37]), but the estimate did not reach statistical significance ($p = 0.094$).

## DISCUSSION

In the present study we investigated the ability of 3 different measures of neurodegeneration to predict the disease course after 1 year in patients suspected of neurodegenerative disease. Evidence of neurodegeneration on both MRI and [18F]FDG -PET in 2 or more lobes compared to having 0-1 lobes affected was associated with a more than fourfold increase in odds of progression after 1 year. Patients with isolated atrophy or hypometabolism-dominated patterns were not at an increased risk of progression after 1 year. These estimates were validated in sensitivity analyses and further confirmed by subgroup analysis of patients with MCI and dementia. The findings indicate that MRI and [18F]FDG -PET holds complementary information which may reflect different aspects of neurodegeneration. To our knowledge, this study is the first to report on the prognostic properties of combined MRI and [18F]FDG -PET quantitative data within a mixed memory clinic cohort.

While we acknowledge that several studies have investigated the relationship between biomarkers of neurodegeneration and progression, most have been conducted in patients with MCI (*Chetelat et al., 2005*; *Fellgiebel et al., 2007*; *Landau et al., 2010*;

**Table 4  Isolated atrophy/hypometabolism patterns and their relation to clinical progression.**

| Variable | Level | Stable | Progressed | Odds ratio (univariable) | Odds ratio (multivariable) |
|---|---|---|---|---|---|
| Age | Mean (SD) | 68.6 (9.7)[c] | 73.6 (9.2)[d] | 1.06 (1.02–1.10, $p = 0.004$) | 1.05 (1.00–1.10, $p = 0.064$) |
| Sex | Female | 50 (50.5) | 27 (54.0) | – | – |
| | Male | 49 (49.5) | 23 (46.0) | 0.87 (0.44–1.72, $p = 0.687$) | 1.21 (0.38–3.87, $p = 0.751$) |
| +Atrophy, -Hypometabolism[a] | 0-1 affected lobes | 58 (58.6) | 32 (64.0) | – | – |
| | 2 or more affected lobes | 41 (41.4) | 18 (36.0) | 0.80 (0.39–1.61, $p = 0.524$) | 1.96 (0.42–9.19, $p = 0.392$) |
| -Atrophy, +Hypometabolism | 0-1 affected lobes | 67 (67.7) | 32 (64.0) | – | – |
| | 2 or more affected lobes | 32 (32.3) | 18 (36.0) | 1.18 (0.58–2.41, $p = 0.654$) | 0.68 (0.16–2.99, $p = 0.612$) |
| +Atrophy, +Hypometabolism | 0-1 affected lobes | 69 (69.7) | 17 (34.0) | – | – |
| | 2 or more affected lobes | 30 (30.3) | 33 (66.0) | 4.46 (2.16–9.22, $p < 0.001$) | 7.60 (1.26–46.01, $p = 0.027$) |
| Frontal isolated atrophy/ hypometabolism | No abnormality | 25 (25.3) | 4 (8.0) | – | – |
| | Any congruence and/or non-isolated atrophy/ hypometabolism[b] | 20 (20.2) | 25 (50.0) | 7.81 (2.33–26.15, $p = 0.001$) | 2.60 (0.36–18.77, $p = 0.344$) |
| | Isolated hypometabolism | 23 (23.2) | 14 (28.0) | 3.80 (1.09–13.24, $p = 0.036$) | 2.54 (0.52–12.48, $p = 0.252$) |
| | Isolated atrophy | 31 (31.3) | 7 (14.0) | 1.41 (0.37–5.37, $p = 0.613$) | 0.87 (0.12–6.46, $p = 0.892$) |
| Temporal isolated atrophy/ hypometabolism | No abnormality | 28 (28.3) | 6 (12.0) | – | – |
| | Any congruence and/or non-isolated atrophy/ hypometabolism | 28 (28.3) | 30 (60.0) | 5.00 (1.80–13.88, $p = 0.002$) | 2.97 (0.42–21.04, $p = 0.276$) |
| | Isolated hypometabolism | 21 (21.2) | 8 (16.0) | 1.78 (0.54–5.90, $p = 0.347$) | 2.00 (0.26–15.51, $p = 0.508$) |
| | Isolated atrophy | 22 (22.2) | 6 (12.0) | 1.27 (0.36–4.50, $p = 0.708$) | 2.43 (0.35–16.91, $p = 0.370$) |
| Parietal isolated atrophy/ hypometabolism | No abnormality | 26 (26.3) | 7 (14.0) | – | – |
| | Any congruence and/or non-isolated atrophy/ hypometabolism | 28 (28.3) | 27 (54.0) | 3.58 (1.33–9.62, $p = 0.011$) | 0.96 (0.16–5.91, $p = 0.967$) |
| | Isolated hypometabolism | 20 (20.2) | 10 (20.0) | 1.86 (0.60–5.74, $p = 0.282$) | 2.06 (0.33–12.99, $p = 0.442$) |
| | Isolated atrophy | 25 (25.3) | 6 (12.0) | 0.89 (0.26–3.02, $p = 0.854$) | 0.76 (0.11–5.19, $p = 0.776$) |
| Occipital isolated atrophy/ hypometabolism | No abnormality | 17 (17.2) | 9 (18.0) | – | – |
| | Any congruence and/or non-isolated atrophy/ hypometabolism | 33 (33.3) | 21 (42.0) | 1.20 (0.45–3.19, $p = 0.712$) | 0.06 (0.01–0.46, $p = 0.006$) |
| | Isolated hypometabolism | 20 (20.2) | 10 (20.0) | 0.94 (0.31–2.86, $p = 0.920$) | 0.29 (0.04–1.84, $p = 0.189$) |
| | Isolated atrophy | 29 (29.3) | 10 (20.0) | 0.65 (0.22–1.92, $p = 0.437$) | 0.25 (0.04–1.64, $p = 0.149$) |

**Notes.**

[a] + refers to a z-score<0 in an affected lobe (right and left hemisphere frontal, temporal, parietal and/or occipital) for either hypometabolism ([18F]FDG-PET uptake) and/or atrophy (MRI volume).

[b] Congruence refers to coexisting hypometabolism ([18F]FDG-PET z-score < 0) and atrophy (MRI z-score < 0) in a specific region (left and/or right hemisphere). Incongruence refers to either [18F]FDG-PET and MRI z-score< 0. Isolation means that the presence of either [18F]FDG-PET or MRI z-score < 0 in a region (frontal, temporal, parietal or occipital) were without the presence of the other. No abnormality = [18F]FDG-PET and MRI z-score>0 for both hemispheres.

[c] n (% of stable) if nothing else stated under level.

[d] n (% of progressed) if nothing else stated under level.

*Dickerson & Wolk, 2013*; *Bouallègue, Mariano-Goulart & Payoux, 2017*; *Altomare et al., 2019*), or AD (*Kester et al., 2009*; *Ottoy et al., 2019*). However, it seems reasonable that biomarkers of neurodegeneration could be used as prognostic tools in all diseases that have neurodegeneration as a prime component of brain pathology. Thus, if a patient is under suspicion of a neurodegenerative disease, regardless of the disease suspected, there seems to be prognostic value in both MRI and [$^{18}$F]FDG -PET quantitative data. Our results are in line with previous findings with regards to the prognostic properties of MRI (*Dickerson & Wolk, 2013*) and [$^{18}$F]FDG -PET (*Mielke et al., 1994*; *Fellgiebel et al., 2007*; *Walhovd et al., 2010a*), although the cohorts in the aforementioned studies differ markedly from ours, as we also included patients with a non-AD dementia and patients with SCD. By doing so, our findings are generalizable to the typical array of patients seen for initial evaluation in a memory clinic. On the other hand, our prognostic markers may also simply be identifying patients with SCD versus patients with MCI or dementia. To overcome the issue of whether our markers were mainly diagnostic or prognostic (i.e., determines the diagnostic categories of MCI and dementia against SCD), we further validated our findings by excluding patients with SCD in a pre-planned analysis and this confirmed our initial results, although we did not attempt to replicate the results in the dementia group only. In analyzing data from this study, we aimed to mimic the clinical approach, with regards to how clinicians may use imaging data, when defining our variables of interest. Investigating larger brain structures such as brain lobes in a quantitative manner might seem more in tune with everyday clinical practice, although not a perfect mimic of it, where the [$^{18}$F]FDG -PET images are visually inspected (*Shivamurthy et al., 2015*; *Gallivanone, Rosa & Castiglioni, 2016*; *Kato et al., 2016*) and thus less emphasis is put on smaller abnormalities, which might drown in unspecific changes. This is in contrast to most other studies investigating [$^{18}$F]FDG -PET in a prognostic setting, where smaller (*Walhovd et al., 2010b*), sometimes unilaterally defined ROIs (*Chetelat et al., 2005*) seem to be the preference. As such, our results show that either MRI or [$^{18}$F]FDG -PET quantitative data with our pre-processing pipeline might have limited prognostic information, but the combination of markers seems to improve the prognostic precision as the markers might highlight different areas of disease pathology, which is also confirmed by studies in AD (*Walhovd et al., 2010a*).

Our model incorporating CSF-total tau could not reliably predict progression, which is in line with other findings (*McGhee et al., 2014*). This could be attributed to the limited number of patients who had CSF-total tau measured. This impaired the statistical power, as the directionality of the point estimates for an increasing number of abnormal markers showed. In a review by *McGhee et al. (2014)* which looked at disease progression markers in AD, the prognostic value of CSF-total tau was small and seemed hard to capture, even in studies including 100 (*Zetterberg et al., 2006*) and 274 (*Andreasen et al., 1999*) patients. Thus, CSF-total tau when measured alone might not contribute in a meaningful manner to assessment of risk of progression but there may be confounding factors limiting the usability of the marker which cannot be adequately accounted for. One aspect that could confound results pertaining to not just tau but possibly any CSF marker, is the blood–brain barrier (BBB) permeability. Studies investigating a surrogate marker of BBB permeability, namely CSF/serum albumin ratio, have shown in Parkinson's disease (*Liguori et al., 2017*), that this

ratio increases with disease severity along with amyloid beta (Aβ) and total tau. Adjusting for this ratio may better elucidate the prognostic value of tau, although conclusions on this matter are merely speculative at this point, as a better understanding of the CSF fluid dynamics and physiology is needed. Also, it has been shown that when combining tau with an AD marker such as Aβ(1-42), cognitive decline can be predicted (*Hansson et al., 2018*), meaning that ratios might be better at predicting progression, although the biological reasoning behind and interpretation of these ratios need to be defined more clearly. Recent studies done using data from the Alzheimer's Disease Neuroimaging Initiative have shown that an increase in CSF-phosphorylated-tau, which is an AD-specific CSF marker, may precede tau-PET positivity (*Meyer et al., 2020*), meaning that an increase in CSF-total-tau could be indicative of early stage disease and thus the risk of progression is lower when only this marker is abnormal.

In a model excluding CSF-total tau as a predictor, we evaluated the prognostic value of a proposed marker of disease progression in AD, namely MRI total ventricular volume. We showed that patients who had either an abnormal value for this marker or [$^{18}$F]FDG -PET or both were at an increased risk of progression, thus confirming in part earlier studies (*Nestor et al., 2008*; *Vemuri et al., 2010*), although the cohorts studied differed. This indicates that this marker might hold prognostic value in a mixed memory clinic cohort. In contrast to this finding, we could not show a statistically significant relationship between total ventricular volume and MMSE change in linear regression models. This may reflect the fact the MMSE perhaps does not capture all aspects of disease progression as viewed by the clinician or simply that MRI ventricular volume cannot stand alone as a prognostic marker.

In an exploratory model we investigated whether the exact lobe-wise anatomical localization of atrophy and/or hypometabolism had an influence on the risk of progression. We found that having intralobal congruent atrophy and hypometabolism and/or non-isolated atrophy/hypometabolism in the occipital lobe was associated with a lower probability of progression compared to having no abnormality. Further, we addressed the concern of multicollinearity by eigensystem analysis. These results could not be replicated in a subgroup of patients with MCI and dementia. This relationship only revealed itself in the multivariable analysis, meaning that the univariable estimate was confounded by other included variables. This would suggest that a subset of patients, maybe especially those with occipital lobe restricted atrophy/hypometabolism somehow deviated in terms of non-occipital neurodegeneration and progression. Our interpretation is that the seemingly protective effect of having atrophy and/or hypometabolism in the occipital lobe is a sign of non-neurodegenerative disease, since the occipital lobe is often spared in neurodegenerative dementias (*Harper et al., 2017*). Patients with isolated occipital atrophy may have a non-neurodegenerative disease (e.g., alcohol related dementia) which may be more stable. Another explanation as to why we see this particular protective effect is that patients are asked to close their eyes during the acquisition of the [$^{18}$F]FDG -PET scan, meaning that if patients are unable to cooperate fully, maybe due to their neurodegenerative disease, this might show up as increased metabolism in the occipital lobe due to visual stimulation. However, the protective effect of this specific pattern also raises the concern

of the validity of the model, and this concern was corroborated by the missing ability to replicate results in subgroup-analyses.

The strengths of this study are the well-described cohort, the clinically defined outcome, and the resemblance of both the cohort and our chosen measures of neurodegeneration to the clinic. This means, that our findings could be generalizable to a broad range of memory clinic patients that want to know their prognosis. Another strength is our clinically-driven hypothesis generation, by which we aimed to emulate the process of visual inspection of MRI and [$^{18}$F]FDG -PET scans a memory clinic physician carries out in clinical practice. Further, this enabled us to convert the images into quantitative data that can be assessed in a statistical manner.

We acknowledge inherent limitations of this study. First and foremost, we chose a pragmatic approach to defining clinical progression. A physician who in most cases knew the patient was given four options of describing the disease course since the diagnosis as perceived by the physician herself, the patient and the caregiver. This non-validated measure of assessing clinical progression might impose bias. We investigated whether a model with MMSE decline of 3 points or more would change the results of our main finding, which it did not, indicating that both measures hold value in assessing progression. A critique of our study could also be that our cohort is too broadly defined and that we did not examine the prognostic value within the etiological diagnoses. On the other hand, markers of neurodegeneration should have usefulness in all neurodegenerative diseases and a relatively high number of patients in clinical practice do not receive a specific etiological diagnosis either because it may not be ascertainable or due to limited resources within centers (*Amjad et al., 2018*). We tried to accommodate this view by choosing larger ROIs that reflect a more general approach to the pathological patterns of neurodegenerative disease, although this limited sensitivity. We are aware of the fact that simply defining atrophy and hypometabolism as presented here done does not necessarily mean that actual atrophy and hypometabolism was present, although it is reasonable to suggest that a large proportion of the studied population indeed have substantial atrophy and hypometabolism due to the diagnoses they eventually received. Nonetheless, this presents a risk of oversimplification. Also, although we employed internal validation to test the robustness of our results, the gold standard of biomarker studies demands external validation of the model in an independent cohort. This was not done and presents a drawback. No control sample was included in the main study, which is why we resort to an intracohort-defined $z$-score. The inclusion of a control sample could have improved the validity of the results, although the selection process associated with inclusion of control samples can suffer from the causes of selection bias and may give false positive findings. We did not consider vascular damage nor did we make amends to filter out the possible effect of vascular damage from our subjects. We believe that this is an area of research that deserves attention, but we also found it to be beyond the scope of this work to account for, as we mainly study the downstream cascades of neurodegeneration and vascular damage can be thought of as an up-stream event. We also did not study the comparison of a clinical progression risk rated at baseline with our method and further, we did not adjust for intracranial volume which could confound our results, although an adjustment for sex

accounts for some of this variation. Further, generalization is limited to patients with a MMSE $\geq$ 18.

A perspective to be added to this study is the relatively new idea of a pre- and post-biomarker counseling effort (*Herukka et al., 2017*), which is applicable for patients diagnosed with MCI. Biomarker counseling should be given as continuous support also after the patient is informed of a positive CSF biomarker result ensuring that the patient is cared for in a way that fits the possibly dire outcome. The results of the present study support the notion that biomarkers indeed hold promise as prognostic markers, but we need to care for and instruct patients that receive such a prognosis in a meaningful, empathetic way, ensuring that patients know what to do to stay healthy for as long as possible. Further, it seems that hypometabolism should also be considered when inspecting a [$^{18}$F]FDG -PET scan even if the MRI shows atrophy, meaning that there probably is excess hypometabolism that is missed when simply ascribing any and all hypometabolism to atrophy.

## CONCLUSIONS

In conclusion, the present findings expand the knowledge on the usefulness of biomarkers of neurodegeneration in the clinical setting. Complementary markers of neurodegeneration add to the prognostic value of the isolated biomarkers. However, these were preliminary findings which need further validation. Future studies with larger cohorts should be conducted in this area to validate our findings and to further elucidate the prognostic value of these biomarkers of neurodegeneration.

## ACKNOWLEDGEMENTS

We would like to extend our thanks to all patients for their willingness to participate in this study.

### Funding

The salary of Mathias Holsey Gramkow was supported by the Research Fund at Rigshospitalet, Copenhagen, Denmark. This work was co-funded by the European Commission under grant agreement 611005 (PredictND). For development of the PredictND tool, VTT Technical Research Center of Finland Ltd has received funding from European Union's Seventh Framework Programme for research, technological development and demonstration under grant agreements 601055 (VPH-DARE@IT), 224328 and 611005. The PredictND consortium consisted of collaborators from VTT Technical Research Centre of Finland, GE Healthcare Ltd, Imperial College London, Alzheimer Europe, Alzheimer Center - VU University Medical Center, Amsterdam, the Netherlands, the Danish Dementia Research Centre, Copenhagen University Hospital, Denmark, the department of Gerontology and Geriatrics of the University of Perugia, 'S. Maria della Misericordia' Hospital of Perugia, Italy, the department of Neurology from the

University of Eastern Finland. The funders had no role in study design, data collection and analysis, decision to publish, or preparation of the manuscript.

### Grant Disclosures

The following grant information was disclosed by the authors:

European Commission under grant agreement 611005 (PredictND).

European Union's Seventh Framework Programme for research, technological development and demonstration: 601055 (VPH-DARE@IT), 224328, 611005.

### Competing Interests

Juha Koikkalainen and Jyrki Lötjönen are shareholders at Combinostics Ltd. No tools from Combinostics Ltd. were directly used in this work.

### Author Contributions

- Mathias Holsey Gramkow conceived and designed the experiments, performed the experiments, analyzed the data, prepared figures and/or tables, authored or reviewed drafts of the paper, and approved the final draft.
- Le Gjerum conceived and designed the experiments, analyzed the data, authored or reviewed drafts of the paper, and approved the final draft.
- Juha Koikkalainen, Jyrki Lötjönen, Ian Law, Steen Gregers Hasselbalch and Gunhild Waldemar analyzed the data, authored or reviewed drafts of the paper, and approved the final draft.
- Kristian Steen Frederiksen conceived and designed the experiments, authored or reviewed drafts of the paper, and approved the final draft.

### Human Ethics

The following information was supplied relating to ethical approvals (i.e., approving body and any reference numbers):

The Scientific Ethics Committee of the Capital Region of Denmark granted ethical approval to carry out the PredictND study (protocol number: H-1-2014-126).

### Data Availability

The anonymized raw data with all variables used in the models are available in a Supplemental File.

### Supplemental Information

Supplemental information for this article can be found online at http://dx.doi.org/10.7717/peerj.9498#supplemental-information.

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
