# Peer review of "Prognostic value of complementary biomarkers of neurodegeneration in a mixed memory clinic cohort"

_PeerJ, doi:10.7717/peerj.9498_

## Round 0.1 · original submission · Major Revisions

I agree with the reviewers' assessments that this is an interesting and well-written article. Although I am issuing a "Major revisions" decision, it appears that most of the reviewer comments could be addressed in a fairly straightforward manner, and I encourage you to revise and resubmit.

Reviewer 1 ·

Basic reporting

This is a well-written manuscript investigating the predictive value of various neurogeneration markers accessible at the memory disorder clinic of tertiary hospitals.
It seems to meet all the standards of this journal including professional article structure, figures, tables, professional English and references.

Experimental design

1. Progression of cognitive function depends on the duration of the disease. You considered age but not disease duration which you might want to control for to investigate the relationship between biomarkers and the clinical progression. Otherwise, it could be a serious confounder.
2. The same is true of the education level which may affect the cognitive decline.
3. The intracohort Z score was used for analysing the biomakrer data in this study. The Z score is conventionally calculated form the mean and standard deviation of the normal control sample which are not available in this study. The readers may be interested to know about the rationale for the intracohort Z score.
4. You mentioned in the inclusion criteria that a T1-weighted MRI with slice thickness>2mm of sufficient quality for analysis. Should it not be slice thickness<2mm rather than >2mm?

Validity of the findings

The way you defined clinical progression is too pragmatic to be taken for granted. You might want to define it using MMSE and CDR scores, for example, 3-4 points decrease on MMSE or an increase on CDR SOB. The decision only based on physician's impression is subjective and most likely will be biased, undermining the validity and also generizability of the findings.

Reviewer 2 ·

Basic reporting

No comment.

Experimental design

Although the methodology is correct, I would like to suggest some points to improve the article

1) Please, specify more about the FDG-PET analysis. What software was used? Is the whole cerebellum used as a reference, just a part of it, or a grey substance?

2) Since we are trying to detect neurodegeneration and not vascular damage, if patients had vascular damage, was any segmentation done to eliminate the effect on the analysis of brain metabolism?

3) About the cerebrospinal fluid (CSF), were Abeta 1-42 and phosphotau not measured? What were the cut-off points for total CSF?

4) Another important aspect, which requires further clarification, is whether the z-score was calculated based on the sample itself. Because if it is the case, it would not be discriminating atrophy versus non-atrophy, but rather patients with more or less atrophy. It would be interesting to use a control group to calculate normality, and afterward a z-score <0 should not be used as an indicator of atrophy, but rather a -1 or -1.5 (1 or 1.5 standard deviations below normality).

Validity of the findings

The final diagnosis of the patients should be specified. Even if it is a mixed cohort, it should be reported if they are patients with Alzheimer's, Frontotemporal Dementia, Lewy, depression, etc. This is an essential aspect because biomarkers should be interpreted in the context of a diagnosis, and not only as positive/negative, atrophy/non-atrophy, hypometabolism/non-hypometabolism. This represents a significant limitation that restricts the generalization of results since not knowing what type of patients we are talking about could affect the interpretation of the tests (e.g., MMSE).

Additional comments

This is an interesting and clinically relevant document on the prognostic properties of atrophy and hypometabolism in patients with suspected neurodegenerative diseases. The authors link several interesting aspects of the field, which gives the article a very solid proposal.

However, there are some considerations that I think the authors should include in a future version before publication to improve the quality of this work. The suggestions were detailed in the specific sections.

---

## Round 0.2 · accepted · Accept

The revised manuscript adequately addresses the reviewers' comments and should make an important contribution to the field.

Reviewer 2 ·

Basic reporting

No comment.

Experimental design

No comment.

Validity of the findings

No comment.

Additional comments

The authors have addressed all the suggestions made in the first review and have provided answers to the questions raised. It is an interesting article that will make an excellent contribution to the field.